# Formulation, Characterization and Evaluation against SH-SY5Y Cells of New Tacrine and Tacrine-MAP Loaded with Lipid Nanoparticles

**DOI:** 10.3390/nano10102089

**Published:** 2020-10-21

**Authors:** Sara Silva, Joana Marto, Lídia Gonçalves, António J. Almeida, Nuno Vale

**Affiliations:** 1OncoPharma Research Group, Center for Health Technology and Services Research (CINTESIS), Rua Dr. Plácido da Costa, 4200-450 Porto, Portugal; saracpsilva21@gmail.com; 2Faculty of Pharmacy, University of Porto, Rua de Jorge Viterbo Ferreira 228, 4050-313 Porto, Portugal; 3Research Institute for Medicines (iMed.ULisboa), Faculdade de Farmácia, Universidade de Lisboa, Avenida Prof. Gama Pinto, 1649-003 Lisboa, Portugal; jmmarto@ff.ulisboa.pt (J.M.); lgoncalves@ff.ulisboa.pt (L.G.); aalmeida@ff.ulisboa.pt (A.J.A.); 4Faculty of Medicine, University of Porto, Al. Prof. Hernâni Monteiro, 4200-319 Porto, Portugal

**Keywords:** tacrine, nanoparticles, SH-SY5Y, MAP, Alzheimer’s disease, drug delivery

## Abstract

Tacrine (TAC) was the first FDA approved drug for the treatment of Alzheimer’s disease, resulting in increased memory and enhanced cognitive symptoms in patients. However, long-term therapy presents poor patient compliance associated with undesired side effects such as nausea, vomiting and hepatoxicity. To improve its therapeutic efficacy and decrease toxicity, the use of nanoparticles could be applied as a possible solution to delivery TAC. In this context, a project has been designed to develop a new nanostructured lipid carrier (NLC) as a delivery system for TAC and conjugate TAC and model amphipathic peptide (MAP) to decrease TAC limitations. Different formulations loaded with TAC and TAC + MAP were prepared using a combination of Compritol 888 ATO as the solid lipid and Transcutol HP as the liquid lipid component. Physical characterization was evaluated in terms of particle size, surface charge, encapsulation efficiency and in vitro drug release studies. Particle size distributions within the nanometer range were obtained with encapsulation efficiencies of 72.4% for the TAC and 85.6% for the TAC + MAP conjugate. Furthermore, cytotoxicity of all NLC formulations was determined against neuroblastoma cell line SH-SY5Y. The optimized TAC delivery system revealed low toxicity suggesting this could be a potential carrier system to deliver TAC. However, TAC + MAP conjugated even encapsulated in the NLC system demonstrated toxicity against the SH-SY5Y cell line.

## 1. Introduction

Tacrine (TAC) is a competitive and reversible inhibitor of both acetylcholinesterase (AChE) and butylcholinesterase [1,2]. TAC was first unsuccessfully investigated to be an antibacterial agent, but further studies revealed analeptic proprieties; therefore, it was used as a morphine antagonist to manage pain in terminal cancer patients. After that, TAC also demonstrated positive effects in patients with anticholinergic delirium so further studies were conducted to evaluate TAC interaction with the cholinergic system [3]. In September of 1993, TAC was approved by the FDA for the treatment of Alzheimer’s disease (AD), being the first approved treatment at the time [4,5]. AD is a progressive neurodegenerative disease and affects almost 6% of world population. Over the years, several studies had demonstrated that AD is a complex disease with multifactorial mechanisms involved. The main mechanisms are amyloid cascade hypothesis [6], hyperphosphorylation of Tau [7], excitotoxicity, oxidative stress, astrocyte impairment and more [8,9,10]. The main AD pathology is associated with widespread neuronal and synaptic loss, particularly of the cholinergic network in the cerebral cortex and hippocampus regions resulting in memory impairment and cognitive and behavior disturbances in patients [2]. The mechanism of action of TAC relies on inhibition of AChE, leading to increased availability of acetylcholine associated with an enhancement on the muscarinic effect, improved memory and cognition of AD patients [2]. However, in long-term treatments, TAC has demonstrated some undesired effects on AD patients such as gastrointestinal effects (nausea, vomiting), shivering and hepatoxicity with an increase of transaminases [11,12]. In addition, pharmacokinetic studies showed that TAC has a short half-life time (around 2 h), low oral bioavailability (17%) and a high clearance rate from systemic circulation [13,14]. Many efforts have been developed to overcome and reduce these side effects [15]. The use of nanoparticles (NP) as a delivery vector can be a promising solution, as they are relatively safe, enhance drug bioavailability, decrease blood protein binding and clearance rate [16,17]. Among the most studied nanoparticulate systems, nanostructured lipid carriers (NLCs) are attracting great scientific interest. In contrast to most polymeric nanoparticle systems, NLCs present a lower cytotoxic risk since their preparation avoids the use of organic solvents, still some sensitivity or irritation can arise from the amount of surfactant added [18]. Moreover, when compared to the common solid lipid NP, NLCs are characterized by a less ordered structure which reflects on higher drug loading capacity and retention during storage. Another great advantage of this delivery system is the ability to functionalize through conjugation, grafting or coating with polymers and specific ligands (aptamer, peptides, antibodies and others) [19]. Thus, the development of upgrade NPs leads to higher stability, improves targeting, increases circulation time and reduces undesired body distribution [19]. Several studies have demonstrated the major advantages of using lipid-based nanoparticles for efficient target and delivery of different compounds against brain diseases [20,21]. NLCs can be prepared with a variety of lipids and surfactants in order to achieve the optimal formulation. In this paper, Compritol 888 ATO^®^ and Transcutol^®^ were used as lipid components and Tween 80 as a surfactant. Compritol 888 ATO^®^ is composed of mono-, di- and triglycerides of behenic acid, is insoluble in aqueous solutions, non-digestible, non-ionic and chemically inert with a high melting point and narrow recrystallization behavior [22]. Transcutol^®^ is diethylene glycol monomethyl ether, usually used as a co-surfactant and has optimal solubilizing proprieties for drugs [23]. Tween 80 was used to stabilize the overall formulation and when used in nanoparticle formulations has demonstrated the ability to enhance blood-brain barrier crossing [24].

Cell-penetrating peptides (CPPs) are mostly used in combination with NPs as targeting ligands, and are facilitating drug delivery through endocytosis or passive transport. One example of efficient drug delivery is the model amphipathic peptide (MAP), presenting 72 h half-life in human serum [25,26] and an enhanced capability to deliver a wide range of compounds such as peptides, siRNA and oligonucleotides [27,28,29]. MAP is a cell penetrating peptide with 18 amino acid residues (KLALKLALKALKAALKLA), presenting repetitive blocks of cationic residue (lysine) and two hydrophobic residues (alanine and leucine), which confer an amphipathic a-helix structure crucial to interact with membrane surfaces and facilitate endocytosis [30]. In this context, CPP can also be used to act as an active compound in different cellular mechanisms. In a previous study, our group demonstrated that, upon conjugation with Rasagiline, MAP could somehow interact with alpha-synuclein aggregates decreasing its formation in a Parkinson’s disease model [31]. Therefore, in multifactorial and complex disorders, such as neurodegenerative diseases, it is important to develop a compound with broad effect and multiple targets. AD is characterized not only by the loss of cholinergic neurons and synapses, but also by the progressive accumulation of protein aggregates in the brain (diffuse and extracellular beta-amyloid plaques and intracellular neurofibrillary tangles accompanied by reactive microgliosis and dystrophic neurites), which causes neuronal destabilization and induces membrane damage and inflammatory response [32,33,34]. 

In this work, we developed a stable and safe dual delivery to improve TAC delivery by combining NLC with TAC or its conjugate with MAP (TAC-MAP) and evaluating their toxicity in vitro, using SH-SY5Y cells to determine its safety for use in future studies of AD models.

## 2. Materials and Methods 

### 2.1. Materials

Tacrine hydrochloride (9-amino-1,2,3,4-tetrahydroacridine hydrochloride hydrate) (TAC) was acquired from Sigma Aldrich (Algés, Portugal). Glyceryl dibehenate (Compritol^®^ 888 ATO) and diethylene glycol monoethyl ether (Transcutol HP^®^) was a kind gift from Gattefossé (France). Polysorbate 80 (Tween 80^®^) was obtained from J. Vaz Pereira S.A. (Benavente, Portugal). The 3-(4,5-dimethyl-2-thiazolyl)-2,5-dipheny-2H-tetrazolium bromide (MTT), sulfarhodamine B sodium (SRB) and phosphate buffer solution (PBS—pH 7.4) were purchased from Sigma-Aldrich (St Louis, MO, USA). Caprylic/capric triglycerides (Miglyol^®^ 812) were a gift from Sasol Olefins & Surfactants GmbH (Hamburg, Germany). Dimethyl sulfoxide (DMSO) was obtained from Merck (Algés, Portugal). For the viability studies, neuroblastoma cell line (SH-SY5Y, ATCC^®^ CRL-2266TM) was used. The culture medium, penicillin-streptomycin was acquired from Sigma-Aldrich and the rest of the supplies were purchased from Millipore Sigma.

### 2.2. TAC-MAP Synthesis

MAP and TAC-MAP were previously developed by our research group. Briefly, MAP was synthetized through Fmoc/tBu solid-phase peptide synthesis (SPPS) methodologies assisted with microwave (MW) energy, using a Liberty Microwave Peptide Synthesizer in conjugation with TAC through “click chemistry”—a classical copper-catalyzed azide−alkyne cycloaddition (CuAAC) reaction.

### 2.3. THA Solubility Studies 

The solubility of TAC was first determined in Compritol 888 ATO, Transcutol HP and Mglyol. Briefly, the solid lipid Compritol 888 ATO was melted at a temperature 10 °C above its melting point (80 °C), in a controlled temperature water bath, while the liquid lipids Miglyol and Transcutol HP were ready to use. Increasing amounts of TAC were successively added, with stirring, until saturation of the lipid was achieved. This occurred when excess of solid TAC persisted for more than 8 h. Each determination was carried out in triplicate (n = 3).

### 2.4. Preparation of NLC

TAC-loaded NLCs were prepared using a modification of a hot high shear homogenization (HSH) method previously described [35]. Briefly, the solid lipid consisting of 300 mg of Compritol 888 ATO was melted at a temperature 10 °C above its melting point. TAC was dissolved in the liquid lipid Transcutol HP (at theoretical concentrations of 20%, 25% and 30%, *w/w*), which was added to the molten solid lipid. A hot aqueous phase consisting of 3% Tween 80 in 10 mL ultra-purified water was added to the lipid phase under high-shear homogenization at 12,3000 rpm for 10 min (Silverson SL2, UK), in a water bath to maintain the temperature. The NLC dispersion was finally allowed to cool in an ice bath with gentle stirring for 5 min. Each formulation was carried out in duplicates (n = 2). The final dispersion was sealed and stored at 4 °C until further use.

### 2.5. Characterization of NLC

#### 2.5.1. Particle Size and Surface Change

Particle size distribution was analyzed by photon correlation spectroscopy using Zetasizer Nano S (Malvern instruments, UK). For this, samples were placed in a specific cuvette and measurements were made at 25 °C. Results were expressed as average particle size and polydispersity index (PI). Zeta potential was calculated through nanoparticle electrophoretic mobility (previously diluted with filtered purified water and placed in an appropriate), using a Zetasizer Nano Z (Malvern instruments, UK). For all measurements, at least three replicate samples were determined.

#### 2.5.2. Encapsulation Efficiency

The entrapment efficiency (EE) and drug loading (DL) in the NLC formulations were determined using an indirect method. The free drug, i.e., unassociated to the NLC, was separated from the particles using centrifugation (Amicon ultra centrifugal filter units; ultra-15, MWCO 100 Kda, Sigma-Aldrich, Algés, Portugal). Briefly, 1 mL of the sample was kept in the upper chamber compartment of the ultra-centrifuge tube and centrifugated at 12,000 rpm for 15 min. The collected sample in the lower chamber was quantified using high-pressure liquid chromatography (HPLC). Entrapment efficiency and drug loading were determined using the following equations:(1)$$\mathrm{Encapsulation}\;\mathrm{efficiency}\;\left(\%w/w\right)=\frac{w1-w2}{w1}\times 100$$
(2)$$\mathrm{Drug}\;\mathrm{loading}\;\left(\%w/w\right)=\frac{w1-w2}{w3}\times 100$$
where w1 corresponds to amount of drug added in the NLCs, w2 is the amount of free drug and w3 is the amount of lipid.

### 2.6. In Vitro Drug Release

#### 2.6.1. TAC and TAC + MAP Release from pH 7.4

Before release studies were performed, NLCs were desalted on Sephadex G-25 medium pre-filled PD-10 columns (GE Healthcare life Science, UK). The release of TAC and TAC-MAP was determined by incubating the nanoparticles in PBS (pH 7.4) with horizontal shaking at 37 °C. At suitable time intervals, each individual sample was centrifuged at 30,000× *g* for 30 min at 4 °C. The amount of TAC and TAC + MAP release was evaluated in the supernatants by HPLC, at 243 nm and 220 nm, respectively.

#### 2.6.2. TAC Release in Human Plasma

The purified NLCs were incubated in a solution of 80% human plasma with horizontal shaking for 48 h at 37 °C. At suitable time intervals, each individual sample was centrifuged 4 times at 13,000× *g* for 10 min at 10 °C. The amount of TAC and TAC-MAP release was evaluated in the supernatants by HPLC at 243 nm and 220 nm, respectively.

### 2.7. In Vitro Cell Viability Studies

The cytotoxicity of all NLC formulations against SH-SY5Y was assessed by 3-(4,5-dimethyl-2-thiazolyl)-2,5-diphenyl-2H-tetrazolium bromide (MTT) assay, Sulforhodamine B (SRB) assay, microscope visualization and cell count.

#### 2.7.1. MTT Assay

For viability determination by the MTT assay, the cells were maintained in culture DMEM supplemented with 10% heat-inactivated fetal bovine serum (FBS) and 1% penicillin and streptomycin at 37 °C with controlled atmosphere of 5% CO2. The day before the experiment, the cells were seeded in a sterile flat bottom 96-well plate at the density of 18,000 cells per well and incubated for 24 h. The medium was then replaced with increasing concentrations of free TAC, free TAC-MAP and drug loaded NLC. Cells were incubated for 24 h, in negative control cells incubated with culture media and sterilized water (5:1) and in positive control, DMSO 10% was added to promote the cell lyses.

After incubation, all media was removed and replaced with 100 μL of MTT solution (0.5 mg/mL). The cells were further incubated for 3 h. After incubation, 100 μL DMSO was added to dissolve the formazan crystals and the absorbance was measured at 570 nm in a microplate reader (Synergy HT, Biotech Instruments inc., USA). All experiments were performed in quadruplicates and the relative cell viability (%) was compared to control cells and was calculated using the following equation:(3)$$Cell\;viability\;\left(\%\;of\;sample\right)=\frac{Abs\;sample}{Abs\;control}\;\times \;100$$

#### 2.7.2. SRB Assay

Cytotoxicity of all formulations was also determined by SRB assay. Using the same conditions and methods described above for MTT assay, after treatment with all NLC formulations, the cells were fixed with 100 μL of ice cold 10% trichloroacetic acid (TCA) per well for 1 h at 4 °C. Then TCA was removed and allowed to dry before adding 100 μL of 0.4% SRB solution. After 1 h, the plates were then washed with running tap water and air dried. The incorporation dye was solubilized by the addition of 200 μL of 10 mM tris buffer per well. The absorbance was measured at 510 nm in a microplate reader. All experiments were performed in quadruplicates using the same equation mentioned above.

#### 2.7.3. Cell Morphology Visualization

After the treatments, cell morphology, growth and cell count were assessed by contrast phase microscope Lionheart FX (Biotech, USA) with use of Gen5 software (Biotech, USA).

### 2.8. Statistical Analysis

Statistical analysis of the experimental data was performed using a one-way analysis of variance (one-way ANOVA) and differences between groups were tested by a one-way ANOVA with GraphPad Prism version 8.0 (GraphPad Software, San Diego, CA). Data were expressed as mean SEM or 95% confidence interval. A *p* < 0.05 value was considered significant. All data are shown as mean SEM.

## 3. Results and Discussion

### 3.1. NLC Characterization

The ability to delivery TAC through nanoparticles presents several advantages over the free drug, with enhanced bioavailability, stability, therapeutic effect and lower toxicity [36,37,38,39,40]. In this context, three different NLC formulations loaded with TAC or TAC-MAP were prepared using Compritol 888 ATO, different amounts of Transcutol HP as liquid lipid, and Tween 80 as the surfactant. Particle size of the NP is crucial to avoid extensive uptake by the mononuclear phagocytic system (MPS), renal elimination rate if too small (less 50 nm) and to allow overcoming the blood–brain barrier. The NLC presented a particle size distribution below 200 nm and polydispersity index (PI) below 0.4 for all cases (see Table 1). Overall, particle size analysis showed that incorporation of TAC and TAC + MAP did not contribute to a significant increase on NLC size when compared with blank NLCs (Figure 1; Table 1). The zeta potential analysis demonstrated negative values of −12.02 to −16.02 mV for Blank NLC and NLC-loaded TAC, whereas, NLC loaded with TAC + MAP, presented a slight increase in ZP, from −4.91 to +9.77 mV possibly due to the presence of the cationic molecules (TAC, MAP and TAC + MAP conjugate).

Preliminary solubility studies demonstrated that TAC had a maximum solubility level in liquid lipid Transcutol HP when compared to the other lipid tested, which clearly affects encapsulation efficiency (EE) and drug loading (DL) [41]. The former (EE) is also a crucial part in developed a delivery system, since higher levels of encapsulation are recommended and reflect on efficacy. For instance, the DL reflects on the amount of drug load per unit weight of the NLC. The EE determined for NLC-loaded with TAC was 72.4%, 56.4% and 58.5% for CT20 formulation, CT25 formulation and CT30 formulation, respectively. The drug loading for these formulations for CT20TAC was 1.60%, CT25TAC was 1.96% and CT30TAC was 3%. Although DL was low, this parameter is not necessarily related to the delivery rate in vivo, as shown by Chu and his team, who demonstrated that nanoparticles with 9% of drug loading were more efficient to deliver higher doses of docetaxel in vivo than nanoparticles containing 20% of drug [42]. For TAC-MAP formulations, EE results demonstrated over 85.0% encapsulation with a drug loading of around 1.46% (see Table 1). The liquid lipid:drug ratio did not affect overall the TAC-NLC particle size and, furthermore, TAC was efficiently encapsulated in all formulations. Additional cytotoxicity experiments were conducted to further optimize the formulations (see below).

### 3.2. In Vitro TAC Release from NLC

Release studies were performed using PBS pH 7.4 and human plasma to take into consideration some parameters that should simulate in vivo conditions, such as pH and plasma components. In terms of pharmacokinetics, TAC presents some drawbacks, such as low bioavailability (17%) and a low half-life (2–3 h) [13]. Therefore, incorporation in NLC can help to overcome these limitations, while providing a slow rate of delivery. Release profiles were obtained for incubation in PBS pH 7.4 for up to 72 h, using sink conditions. After 24 h, almost 50.8% of TAC was released, 56.0% in 48 h and 60.7% in 72 h. The rapid initial release of TAC is possible due to the molecules of TAC present at the surface of the NLC. We can also observe that in the first 30 min there was an initial burst effect followed by a steady release up to 72 h, which can be attributed to the diffusion of TAC from the core of the NLC (Figure 2B). In the case of TAC-MAP formulations, incubations in PBS for 24 h did not show any release of content. One possible explanation is the aggregation capacity of MAP to form a β-sheet when interacting with lipid bilayers in order to fulfill its penetrating function efficiently [43]. The hydrophobic residues of MAP interact with lipid acyl chains through Van der Walls interactions and can form H-bonds of β-sheet structure, all of these factors could stabilize the overall system [44,45]. The results obtained with incubation in plasma were a release of 42.30% in 24 h incubation (Figure 2B). The formulations with TAC-MAP showed that no release occurs.

### 3.3. In Vitro Cell Viability Studies of TAC and TAC-MAP

In vitro cytotoxicity studies are important to validate the safety of the delivery system. With this purpose, toxicity was assessed by measuring the cell viability of the neuroblastoma cell line (SH- SY5Y), based on its ability to differentiate in a model of cholinergic neurons, treated with all formulations developed using the MTT and SRB assays.

#### 3.3.1. MTT Assay

The MTT assay provides information about cell metabolism activity and is frequently used. The reduction of cellular metabolic activity has been used as an indicator of compromised cells. In this work, all formulations were tested for a period of 24 h in a concentration range of 1 μM to 25 μM. Results show that in CT25 formulations up to 20 μM, with or without TAC, no evidence of acute cytotoxicity was observed with an 80% of cell viability (Figure 3). The formulation of CT30 was relatively safe at 10 μM with a decrease in 20% of cell viability (Figure 3).

On the other hand, CT20 formulations presented a high level of cytotoxicity in SH-SY5Y cells. One possible explanation for this result can be the higher PI, which may lead to a tendency to aggregate and increase toxicity. The slight decrease of 20% of cell viability was expected due to the use of Tween 80 which is a surfactant considered to present some toxicity [35]. The CT20 formulation showed to affect cell viability (Figure 3).

The TAC + MAP control demonstrated substantial toxicity, while TAC-MAP alone showed toxic activity at 2.5 μM with 50% viability reduction (Figure 4). MAP was incorporated in the NLC formulation to further increase TAC delivery. In fact, MAP incorporation increased delivery efficiency of overall formulations, with a slight increase of toxicity when compared with free TAC + MAP (Figure 4). Still, at 2.5 μM, CT25 demonstrated lower toxicity levels and CT20 presented higher toxicity levels when compared with other formulations.

#### 3.3.2. SRB Assay

In addition, SRB assay was used to corroborate the MTT results, being a good indicator of cell density determination based on protein content. Unlike MTT, the SRB method does not allow distinguishing between viable and dead cells, but it can be used to detect cytotoxic effects of a drug [46,47]. The SRB assay results obtained corroborate those from the MTT. Formulations CT25 and CT30 were not significantly cytotoxic in the same concentration range and CT20 formulations, either blank or TAC-loaded, were more toxic when compared to control (Figure 5). The TAC + MAP treatment and TAC + MAP encapsulated NLC demonstrated strong toxicity against SH-S5Y5 at 2.5 μM with significant cell viability reduction similar to the results obtain with MTT assay.

### 3.4. Cell Morphology Visualization

The morphology of neuroblastoma SH-SY5Y cell line was evaluated after 24 h incubation with all formulations (Figure 5 and Figure 6). This microscope cellular visualization was performed by Lionheart FX and its crucial to corroborate the data obtained with MTT and SRB assays. In the treatment with tacrine, CT25 B, CT30 B, CT25TAC and CT30 TAC (10 μM), we can observe a small decrease of cell viability but no substantial cell morphology differences, thus supporting the in vitro cytotoxicity data (Figure 6). For instance, in the treatment with CT20 B and CT20TAC we observed an increase in cell number and small morphology differences which supports the in vitro cytotoxicity data (Figure 6). The cell count analysis corroborated with cell visualization and demonstrated that treatment with tacrine and all NLC formulations did not affect the number of cells in culture up to 5 μM (Figure 7). Over 10 μM, the treatment with CT20 B, CT30 B, CT20TAC and CT30TAC resulted in a decrease of cell count (Figure 7).

Treatment with the CPP MAP conjugated with TAC and NLCs encapsulated with TAC + MAP had a loss of cellular conformation with the reduction of the prolongations and rounding of the cells, indicating cell death for the concentration of 2.5 μM, as expected (Figure 8). For the lower concentration, 1 μM treatment with CT25TAC + MAP and CT30TAC + MAP resulted in a slight decrease in cell viability and no SH-SY5Y cell morphology differences, thus supporting the in vitro cytotoxicity assays (Figure 8). The cell count analysis corroborated with cell visualization and demonstrated that NLCs loaded with TAC-MAP affect the number of cells in culture at low concentration (2.5 μM) (Figure 9).

## 4. Conclusions

Stable NLC formulations were prepared revealing well-determined size distribution and high encapsulation efficiency for TAC and TAC-MAP. Moreover, cell viability studies conducted demonstrated that NLC encapsulated with TAC were well tolerated and safe against SH-SY5Y cells. In case of TAC-MAP-loaded NLCs, we observed for the concentrations tested high toxicity and accentuated decrease of cell viability. TAC-MAP in previous studies demonstrated high toxicity in high concentration. In the future, it will be interesting to evaluate toxicity and elucidate biodistribution profiles of TAC-loaded NLCs in an in vivo model of AD, particularly to determine protein aggregation levels of ß-amyloid after administration of low concentration TAC-MAP-loaded NLC.

## Figures and Tables

**Figure 1 nanomaterials-10-02089-f001:**
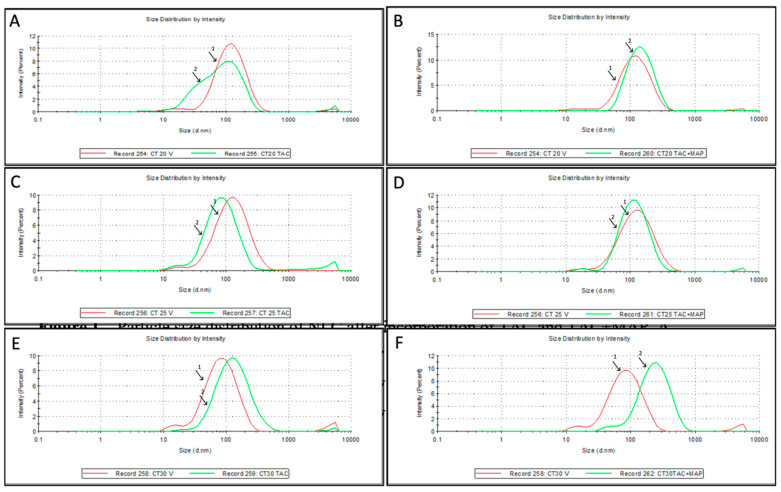
Particle size distribution of NLC after incorporation of TAC and TAC + MAP. (**A**) CT20 V (blank-1) vs. CT20 TAC (2); (**B**) CT20 V (blank-1) vs. CT20 TAC + MAP (2); (**C**) CT25 V (blank-1) vs. CT25 TAC (2); (**D**) CT25 V (blank-1) vs. CT25 TAC + MAP (2); (**E**) CT30 V (blank-1) vs. CT30 TAC (2); (**F**) CT30 V (blank-1) vs. CT30 TAC + MAP (2).

**Figure 2 nanomaterials-10-02089-f002:**
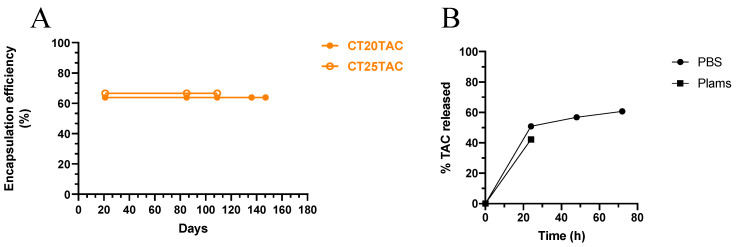
(**A**) Stability of encapsulated TAC in the formulations after 144 days in storage 4 °C. (**B**) Release profiles of TAC from Compritol 888 ATO NLC in PBS, pH 7.4 and 37 °C.

**Figure 3 nanomaterials-10-02089-f003:**
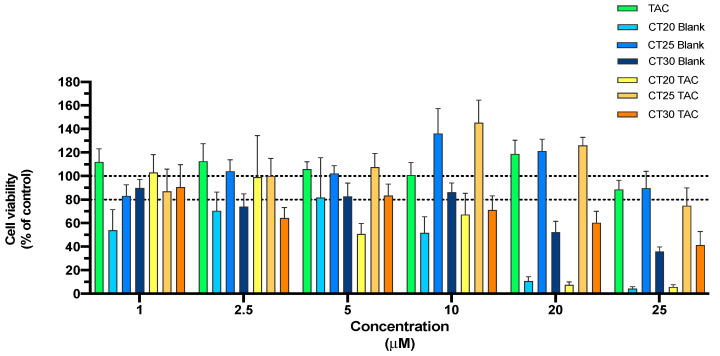
Relative cell viability of SH-SY5Y cell line measured by the MTT reduction after 24 h incubation. Treatment with free TAC, blank NLC and NLC loaded with TAC (1–25 µM). Results are expressed as mean SEM (n = 4). Statistical analysis between the control group and other groups was performed using one-way ANOVA with Dunnet’s post hoc test (* *p* < 0.05).

**Figure 4 nanomaterials-10-02089-f004:**
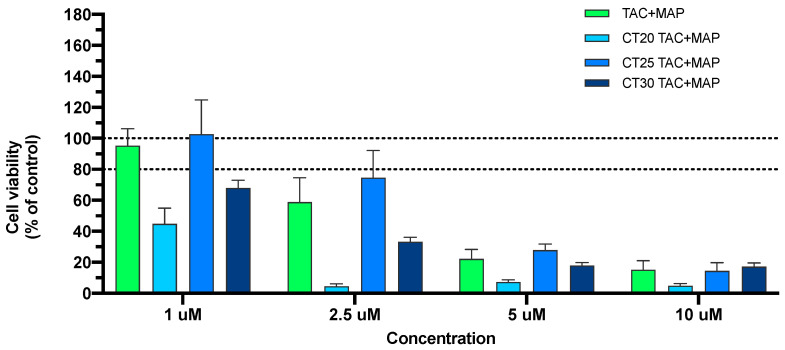
Relative cell viability of SH-SY5Y cell line measured by the MTT reduction after 24 h incubation. Treatment with free TAC-MAP and NLC loaded with TAC-MAP (1–10 μM). Results are expressed as mean SEM (n = 4). Statistical analysis between the control group and other groups was performed using one-way ANOVA with Dunnet’s post hoc test (* *p* < 0.05).

**Figure 5 nanomaterials-10-02089-f005:**
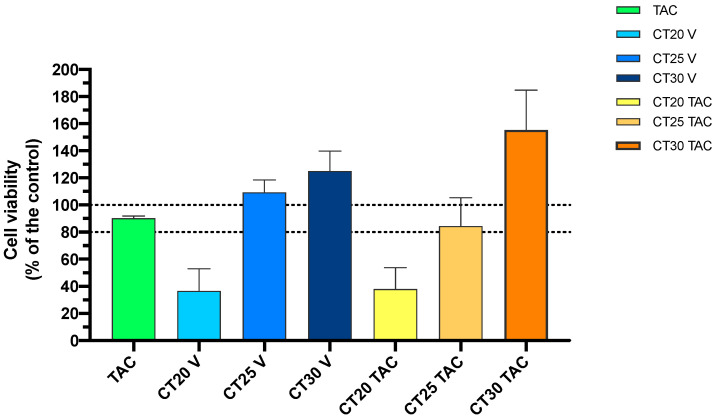
Relative cell viability of SH-SY5Y cell line measured by the SRB assay after 24 h incubation. Treatment with free TAC, blank NLC and NLC loaded with TAC at 10 µM. Results are expressed as mean SEM (n = 4). Statistical analysis between the control group and other groups was performed using one-way ANOVA with Dunnet’s post hoc test (* *p* < 0.05).

**Figure 6 nanomaterials-10-02089-f006:**
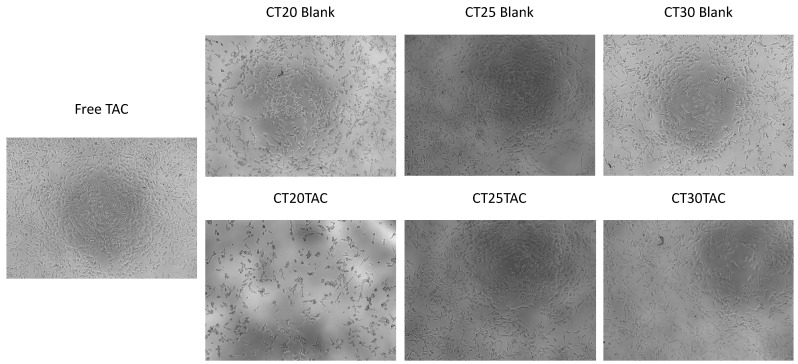
Microscope cellular visualization through Lionheart FX of NLCs blank and with TAC for a concentration of 10 μM.

**Figure 7 nanomaterials-10-02089-f007:**
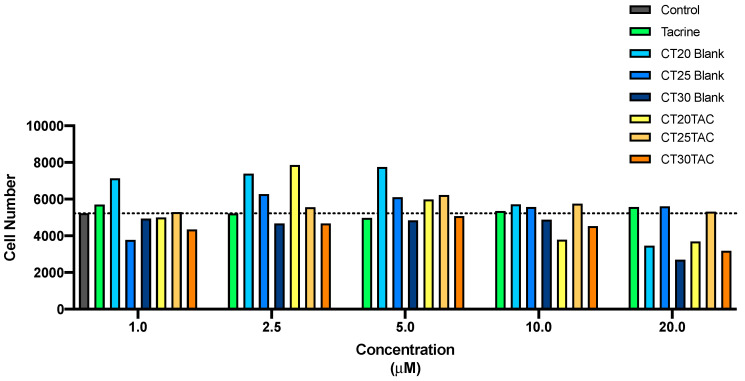
Cell count through Lionheart FX treatment with free TAC, blank NLC and NLC loaded with TAC (1–25 µM) for 24 h incubation.

**Figure 8 nanomaterials-10-02089-f008:**
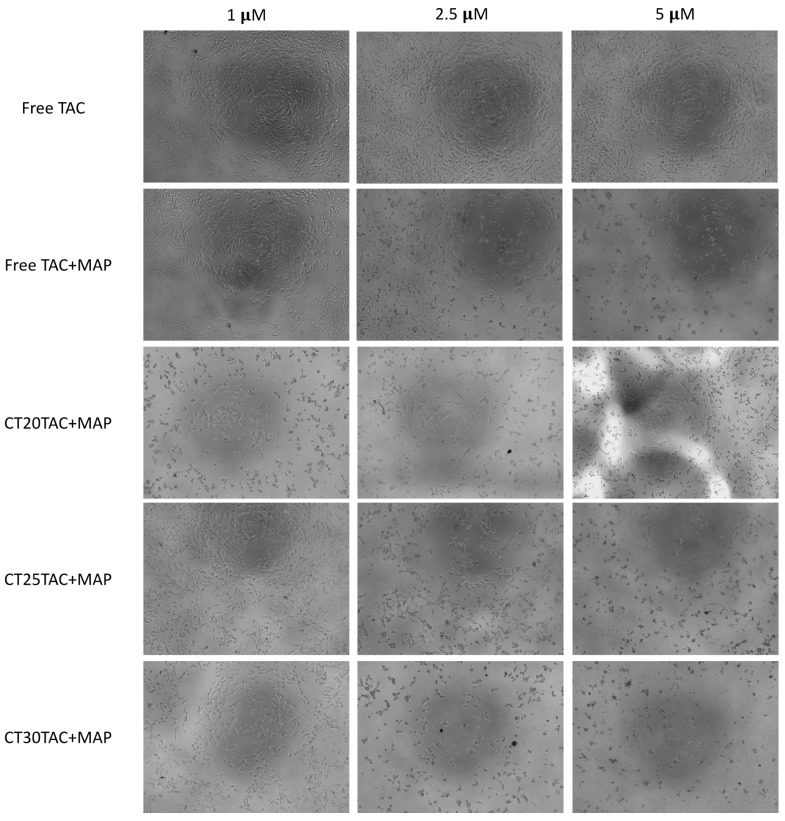
Microscope cellular visualization through Lionheart LX of NLCs loaded with TAC-MAP in concentrations of 1 μM, 2.5 μM and 5 μM.

**Figure 9 nanomaterials-10-02089-f009:**
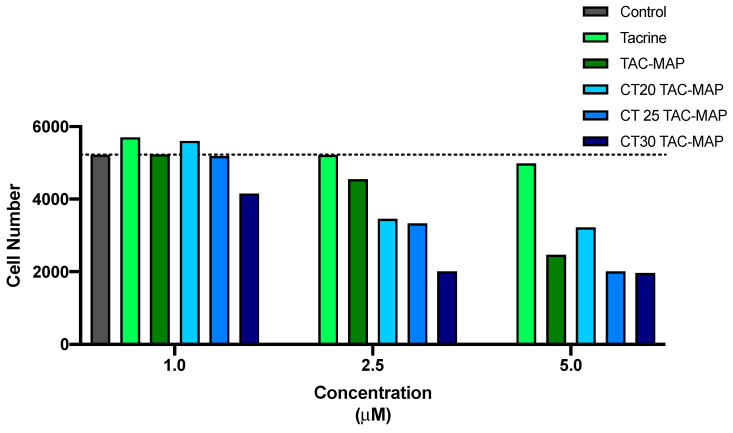
Cell count through Lionheart FX of NLCs loaded with TAC-MAP for concentrations of 1, 2.5 and 5 μM.

**Table 1 nanomaterials-10-02089-t001:** Physicochemical properties of: empty nanostructured lipid carrier (NLC), tacrine (TAC)-loaded NLC, TAC-model amphipathic peptide (MAP)-loaded NLC freshly (mean SD, n = 2).

NLC	NLC composition	PS (nm)	PI	ZP (mV)	EE%	DL%
**CT20 B**	Lipids: 80% C and 20% TE Surfactant: 3% T80	96.995 ± 15	0.2947 ± 0.068	−13.589 ± 1.1	--	--
**CT25 B**	Lipids: 75% C and 25% TE Surfactant: 3% T80	111.78 ± 35	0.2547 ± 0.003	−12.017 ± 0.5	--	--
**CT30 B**	Lipids: 70% C and 30% TE Surfactant: 3% T80	71.465 ± 5	0.3535 ± 0.017	−14.633 ± 0.4	--	--
**CT20TAC**	Lipids: 80% C and 20% TE Surfactant: 3% T80	80.415 ± 9	0.410 ± 0.023	−16.021 ± 1.0	72.4 ± 10.3	1.60 ± 0.29
**CT25TAC**	Lipids: 75% C and 25% TE Surfactant: 3% T80	71.857 ± 12	0.365 ± 0.884	−13.484 ± 0.02	56.4 ± 7.8	1.96 ± 0.11
**CT30TAC**	Lipids: 70% C and 30% TE Surfactant: 3% T80	123.733 ± 24	0.290 ± 0.013	−12.250 ± 1.0	58.5 ± 7.0	3.00 ± 0.43
**CT20TAC + MAP**	Lipids: 80% C and 20% TE Surfactant: 3% T80	281.634 ± 206	0.290 ± 0.093	−9.774 ± 5.8	>85.2	>1.23
**CT25TAC + MAP**	Lipids: 75% C and 25% TE Surfactant: 3% T80	101.029 ± 13	0.280 ± 0.021	−8.194 ± 3.4	>85.2	>1.33
**CT30TAC + MAP**	Lipids: 70% C and 30% TE Surfactant: 3% T80	189.284 ± 207	0.252 ± 0.004	−4.908 ± 0.4	>85.2	>1.81

PS: Mean particle size; PI: polydispersity index; EE: encapsulation efficiency; DL: drug loading; C: Compritol ATO 888; TE: Transcutol HP; T80: Tween 80; B: Blank.

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
