# Peer review of "Formulation, Characterization and Evaluation against SH-SY5Y Cells of New Tacrine and Tacrine-MAP Loaded with Lipid Nanoparticles"

_nanomaterials, 2020, doi:10.3390/nano10102089_

Round 1

Reviewer 1 Report

Vale and co-workers report in this work a new nanoformulation for delivery of Tacrine, a drug to treat Alzheimer disease. Tacrine has undesired effects on AD patients such as gastrointestinal effects (nausea, vomiting), shivering and hepatoxicity with an increase of transaminases and therefore, a new nanoformulation is highly beneficial for patients. Physical characterizations including particle size, surface charge, encapsulation efficiency and in vitro drug release studies have been investigated in details and very well. Importantly, the novel nanoformulation is relatively well-tolerated and thus, is a promising nanocarrier system to deliver Tacrine. Overall, the work is novel, well-conducted and has the potential to improve the treatment of Alzheimer disease. As such, I strongly recommend publishing on Nanomaterials after minor revisions.

  1. In the introduction, it is useful to have some sentences introducing the lipids used in this work (ATO 888, Transcutol and Tween 80).
  2. The sentence “the use of nanoparticles (NP) as a delivery vector can be a promising solution, as they are relatively safe, enhance drug bioavailability, decrease blood protein binding and clearance rate” should cite another relevant reference (Small 2018, 10.1002/smll.201801702) to support the statement.
  3. The authors should also cite a relevant work on both polymeric nanoparticle and nanostructured lipid carriers (Small 2020, 10.1002/smll.202002861) for this sentence “In contrast to most polymeric nanoparticle systems, NLC present a lower cytotoxic risk since their preparation avoids the use of organic solvents, still some sensitivity or irritation can arise from the amount of surfactant added.”
  4. Is it possible to provide Cryo-TEM to show the morphology of the nanostructured lipid carriers?
  5. In table 1, data of drug loading are quite low (CT20TAC was 1.60%, CT25TAC was 1.96% and CT30TAC was 3%) while the encapsulation efficiency was high. The authors may elaborate on this point more in the text.
  6. Figure 2 shows the particles were very stable after 144 days in storage 4 °C but rapid release after 20h in PBS, pH 7.4 and 37 °C. What was the trigger for the drug release? The authors may discuss the mechanism of release in the text.
  7. It is worthwhile to have a scheme showing structures of the lipids and nanoparticles formed and release drugs under certain conditions.
  8. Data in figure 4 shows that CT20 has the most profound effect on cell viability of SH-SY5Y cell line. The authors may discuss this interesting result further.

Author Response

Reviewer 1:

Comments and Suggestions for Authors

Vale and co-workers report in this work a new nanoformulation for delivery of Tacrine, a drug to treat Alzheimer disease. Tacrine has undesired effects on AD patients such as gastrointestinal effects (nausea, vomiting), shivering and hepatoxicity with an increase of transaminases and therefore, a new nanoformulation is highly beneficial for patients. Physical characterizations including particle size, surface charge, encapsulation efficiency and in vitro drug release studies have been investigated in details and very well. Importantly, the novel nanoformulation is relatively well-tolerated and thus, is a promising nanocarrier system to deliver Tacrine. Overall, the work is novel, well-conducted and has the potential to improve the treatment of Alzheimer disease. As such, I strongly recommend publishing on Nanomaterials after minor revisions.

We thank the reviewer for the observations; we also acknowledge the time spent in the review and the perceptive comments that will undoubtedly improve the quality of the manuscript. The response is addressed below.

  1. In the introduction, it is useful to have some sentences introducing the lipids used in this work (ATO 888, Transcutol and Tween 80).

Thank you for the suggestion. We added to the introduction few sentences introducing the lipids and surfactant used.

  1. The sentence “the use of nanoparticles (NP) as a delivery vector can be a promising solution, as they are relatively safe, enhance drug bioavailability, decrease blood protein binding and clearance rate” should cite another relevant reference (Small 2018, 10.1002/smll.201801702) to support the statement.

This relevant reference has been added. Thank you.

  1. The authors should also cite a relevant work on both polymeric nanoparticle and nanostructured lipid carriers (Small 2020, 10.1002/smll.202002861) for this sentence “In contrast to most polymeric nanoparticle systems, NLC present a lower cytotoxic risk since their preparation avoids the use of organic solvents, still some sensitivity or irritation can arise from the amount of surfactant added.”

This relevant reference has been added. Thank you.  

  1. Is it possible to provide Cryo-TEM to show the morphology of the nanostructured lipid carriers?

We recognize the importance of that experiment, but we didn’t conduct that analysis.

  1. In table 1, data of drug loading are quite low (CT20TAC was 1.60%, CT25TAC was 1.96% and CT30TAC was 3%) while the encapsulation efficiency was high. The authors may elaborate on this point more in the text.

Thank you for the pertinent comment. This issue is now discussed in the text.

  1. Figure 2 shows the particles were very stable after 144 days in storage 4 °C but rapid release after 20h in PBS, pH 7.4 and 37 °C. What was the trigger for the drug release? The authors may discuss the mechanism of release in the text.

Thank you for the pertinent question. Figure 2 demonstrates that our NLC formulation is stable and no expulsion of drug occur in the period of 144 days. Incubation in PBS, pH7.4, lead to steady release of TAC, which occurs probably due to the change in pH and temperature that influence the transition temperatute of the lipid networks and allow faster diffusion of the drug out of the nanocarrier.

  1. It is worthwhile to have a scheme showing structures of the lipids and nanoparticles formed and release drugs under certain conditions.

Thank you for the pertinent suggestion. However we do not see any benefit of adding so many schemes, taking into consideration the amount of images that it already presents.

  1. Data in figure 4 shows that CT20 has the most profound effect on cell viability of SH-SY5Y cell line. The authors may discuss this interesting result further.

Thank you for the suggestion. This result is now further discussed in the text.

Reviewer 2 Report

It seems the authors were aiming to improve the limitations of tacrine, its therapeutic efficacy and toxicity. They tried to address the toxicity issue by nanoparticle-based formulation against a single cell line (SH-SY5Y cells). The physicochemical characterizations of the nanoparticles were done. The sizes of the particles were less than 200 nm with polydispersity index of between 0.252-0.41. The zeta potentials were also determined and were less than -20mV.  Even though the size range (less than 200nm) seems to be good and PDI shows the particles are not monodispersed. Moreover, the zeta potential that tells about the stability of the formulations (particles) show less than -20mV and show the particles are not stable in the formulation. However, the authors concluded that they able to prepar Stable NLC formulations but the data do not show that. The Zeta potential results suggest the formulation are most likely temporary and are not stable.

Over all, I do not see novelty in the formulations.  The toxicity issue was not well addressed, and the authors are recommended to test the different formulations (nanoparticles) as compared to equal amount of their respective free drug and thus they can tell how much the formulation decreased the toxicity of the drug (tacrine).  The writing is not well organized and need major re-writing in such a way it clearly shows their research questions and how the results dressed their research questions.   

Author Response

Reviewer 2:

Comments and Suggestions for Authors

It seems the authors were aiming to improve the limitations of tacrine, its therapeutic efficacy and toxicity. They tried to address the toxicity issue by nanoparticle-based formulation against a single cell line (SH-SY5Y cells). The physicochemical characterizations of the nanoparticles were done. The sizes of the particles were less than 200 nm with polydispersity index of between 0.252-0.41. The zeta potentials were also determined and were less than -20mV.  Even though the size range (less than 200nm) seems to be good and PDI shows the particles are not monodispersed. Moreover, the zeta potential that tells about the stability of the formulations (particles) show less than -20mV and show the particles are not stable in the formulation. However, the authors concluded that they able to prepar Stable NLC formulations but the data do not show that. The Zeta potential results suggest the formulation are most likely temporary and are not stable.

Over all, I do not see novelty in the formulations.  The toxicity issue was not well addressed, and the authors are recommended to test the different formulations (nanoparticles) as compared to equal amount of their respective free drug and thus they can tell how much the formulation decreased the toxicity of the drug (tacrine).  The writing is not well organized and need major re-writing in such a way it clearly shows their research questions and how the results dressed their research questions.  

We thank the reviewer for the all observations and also acknowledge the time spent in the review to contribute for the increase quality of the manuscript. In firstly addressing the physical characterization of the NLC in the results, we assume this article as a preliminary study to achieve the optimal new NLC formulation capable to efficiently encapsulate TAC and in the future be further modified with a CPP and in the process evaluated its toxicity against SH-SY5Y, to ensure nanocarrier safety. The results express the data obtained for the all the analysis conducted and even with low zeta potential we could demonstrate the NLC were capable to ensure TAC encapsulation for 144 days and low cell viability decrease on the CT25 and CT30 TAC formulations.

Round 2

Reviewer 2 Report

None of my original review comments were fully addressed but the paper has of interest. Therefore, I do not want the manuscript being rejection but wanted the authors to improve the manuscript by adding some more experiments. 

Author Response

Response to the reviewers’ comments

Nanomaterials

Reference: nanomaterials-935959

Title: Formulation, characterization and evaluation against SH-SY5Y cells of new tacrine and tarine-MAP loaded with lipid nanoparticles

Sara Silva, Joana Marto, Lídia M. Gonçalves, António José Almeida, Nuno Vale

Reviewers' comments:

Reviewer 2:

Comments and Suggestions for Authors

None of my original review comments were fully addressed but the paper has of interest. Therefore, I do not want the manuscript being rejection but wanted the authors to improve the manuscript by adding some more experiments.

We thank the reviewer for the observations; we also acknowledge the time spent in the review and the perceptive comments that will undoubtedly improve the quality of the manuscript. The response is addressed below and manuscript was edited at red.

In this new version of the manuscript 3 new figures have been added (see below) in order to make the work more robust.

[Figure 5]

Figure 5 - Relative cell viability of SH-SY5Y cell line measured by the SRB assay after 24h incubation. Treatment with free TAC, blank NLC and NLC loaded with TAC at 10 µM. Results are expressed as mean SEM (n = 4). Statistical analysis between the control group and other groups was performed using one-way ANOVA with Dunnet’s post hoc test (*p<0.05).

[Figure 7]

Figure 7 - Cell count through Lionheart FX treatment with free TAC, blank NLC and NLC loaded with TAC (1-25 µM) for 24h incubation.

[Figure 9]

Figure 9 - Cell count through Lionheart FX of NLCs loaded with TAC-MAP for concentration of 1, 2.5 and 5 μM.
